# OpenReview forum: "FRABench and UFEval: Unified Fine-grained Evaluation with Task and Aspect Generalization"
_ICLR.cc/2026/Conference — ICLR 2026 Oral_

### Official Review · Reviewer_vDBt · 2025-10-25

**Soundness:** 2
**Presentation:** 3
**Contribution:** 3
**Rating:** 4
**Confidence:** 3

**Summary:**

This paper proposes UFEval, a unified fine-grained evaluator designed to achieve task and aspect generalization for MLLMs. To support this, the authors build a hierarchical taxonomy with 112 distinct aspects across four task types, and based on this, construct FRABench, a large-scale, multi-modal, aspect-level evaluation benchmark, obtained through a mix of human and GPT-4o annotations. They then train UFEval on FRABench and evaluate its generalization ability through in-domain and out-of-domain experiments, demonstrate UFEval’s strong performance as an evaluator compared with baselines, and explore its application in preference alignment for both image understanding and image generation tasks.

**Strengths:**

1. The work introduces UFEval, the first unified fine-grained evaluator designed with both task and aspect generalization capabilities. It covers four diverse evaluation tasks: Natural Language Generation (NLG), Image Understanding (IU), Image Generation (IG), and Interleaved Text-Image Generation (ITIG). Complementary to this, the paper constructs FRABench, a large-scale, multi-modal, and aspect-level comprehensive evaluation dataset, which directly addresses the critical lack of resources required for training such a universal evaluator.

2. The trained UFEval evaluator demonstrates good performance across various benchmarks.

3. The study shows a substantial amount of work and extensive experiments.

**Weaknesses:**

1. While the paper proposes an ambitious taxonomy covering 112 distinct aspects, there are potential weaknesses in how this taxonomy is constructed. The authors note that they directly adopted existing aspect tree structures from previous studies and incorporated additional aspects "based on their definitions". However, the methodology for reconciling differences or conflicts between these source taxonomies is not clearly articulated. Moreover, the paper provides no evidence of expert validation, inter-rater agreement, or any form of consistency checking to support the robustness of the taxonomy.

2. The inclusion of 112 aspects also raises concerns about semantic redundancy. Many aspects seem to overlap substantially in meaning, making them difficult to distinguish in practice. For example:

   - Accuracy and Instruction Following are strongly correlated: when a model accurately completes a task, it typically also follows instructions, and deviations from instructions often result in inaccuracy.

   - Creativity-related aspects such as Appeal, Engagingness, and Creativity appear to measure very similar qualities and may not represent truly distinct evaluative dimensions.

   The paper argues that the FRA-OOD test set contains "unseen task-specific aspects" and uses this as evidence of UFEval's generalization ability. However, if these "unseen" aspects are semantically overlapping or near-synonymous with training aspects, the evaluator may not be demonstrating genuine generalization. Instead, it may simply be transferring learned evaluation criteria to synonymous labels, which is conceptually weaker than true out-of-domain generalization. This issue undermines the validity of the claimed aspect-level generalization.

3. The paper’s core claim is that jointly learning to assess multiple visual aspects and tasks leads to synergistic effects. To support this, the authors compare models trained on a single task (e.g., “w/ IU”) with the multi-task model ("Ours"), and report performance improvements in Tables 4 and 5. However, the "Ours" model is trained on a larger and more diverse dataset than the other ablations. This makes it difficult to disentangle whether the observed gains actually stem from cross-task synergy or simply from exposure to more varied training data. Without a controlled experiment where the total training data volume is held constant across conditions, the evidence provided does not conclusively support the claimed synergistic effect. A data-controlled ablation would strengthen the argument for true multi-task synergy.

**Questions:**

1. What specific methodology was used to reconcile conflicts between the source taxonomies?

2. What empirical evidence (e.g., expert validation) can be provided to support the robustness and distinctness of the 112 aspects?

3. Given the high semantic overlap among many aspects, how can the authors prove that the performance on "unseen" aspects represents true generalization to novel concepts, and not just inference on synonymous labels?

---

> ### Author Response · Authors · 2025-11-21
> **Response to reviewer vDBt (1/5)**
>
> Dear Reviewer vDBt,
>
> We appreciate your time and constructive feedback, which has been invaluable in improving our work. We have taken your concerns very seriously and have made every effort to address your questions and incorporate your suggestions.
>
> >W1: While the paper proposes an ambitious taxonomy covering 112 distinct aspects, there are potential weaknesses in how this taxonomy is constructed. The authors note that they directly adopted existing aspect tree structures from previous studies and incorporated additional aspects "based on their definitions". However, the methodology for reconciling differences or conflicts between these source taxonomies is not clearly articulated. Moreover, the paper provides no evidence of expert validation, inter-rater agreement, or any form of consistency checking to support the robustness of the taxonomy.
>
> >Q1: What specific methodology was used to reconcile conflicts between the source taxonomies?
>
> A: Thank you for your comment regarding “the methodology for reconciling differences or conflicts between these source taxonomies is not clearly articulated”. We acknowledge that our initial description may have been insufficient, leading to potential ambiguity. To address this, we have added a detailed explanation in the revised manuscript (Line 159) under the section **Aspect Tree Construction**.
>
> **Aspect Tree Construction**.
>
> 1. Adoption of Established Taxonomies
>
> To ensure consistency and robustness, we prioritized aspect trees from peer-reviewed literature. Specifically, we adopted the following structures for Readability [1], Bias [2], Instruction Following [3], Hallucination [4], Alignment [5], Complement [6], Image Coherence [7], Image Quality [8], Semantic Consistency [9], and Toxicity [10]. Relying on these published works ensures the validity and rationality of the foundational structure.
>
> **Clarification on Conflicts:** As these source taxonomies span distinct dimensions without overlapping components, no structural conflicts arose during integration.
>
> 2. Integration of Additional Aspects
>
> For the remaining aspects without an established hierarchical structure, we handle them as follows:
>
> - Bidirectional Matching Strategy: First, we check whether the remaining aspect's name appears within the definition of the existing tree's root node. If a match is found, we recursively traverse the child nodes to identify the most specific insertion point and append the aspect as a child of the last matched node. Conversely, if no downward match is found, we verify whether the root node's name appears within the definition of the aspect. In this case, the aspect implies a broader concept and is inserted as the parent of the current root. This applies to the following aspects: [Clarity in Readability], [Semantic Consistency in Image Coherence], and [Integration, Complement, and Alignment in the Text-Image Relationship].
> - Creation of new root nodes: For aspects that do not match any existing tree, we established them as independent root nodes to minimize subjective bias and avoid misleading categorizations. These nodes correspond to all green peer-level nodes within the TAs tree in Figure 1 (Line 162).
>
> [1]Are llm-based evaluators confusing nlg quality criteria? (ACL)
>
> [2]Measuring bias in ai models: an statistical approach introducing n-sigma. (COMPSAC)
>
> [3]Evaluating large language models at evaluating instruction following. (ICLR)
>
> [4]A survey on hallucination in large language models: Principles, taxonomy, challenges, and open questions. (TOIS)
>
> [5]T2i-compbench: A comprehensive benchmark for open-world compositional text-to-image generation. (NIPS)
>
> [6]Categorizing and Inferring the Relationship between the Text and Image of Twitter Posts. (ACL)
>
> [7]Holistic Evaluation for Interleaved Text-and-Image Generation. (ACL)
>
> [8]Q-Eval-100K: Evaluating Visual Quality and Alignment Level for Text-to-Vision Content (CVPR)
>
> [9]Neural Storyboard Artist: Visualizing Stories with Coherent Image Sequence. (MM)
>
> [10]RealToxicityPrompts: Evaluating Neural Toxic Degeneration in Language Models. (EMNLP)
>
> **Related Revised Section: 3.1.2 Aspect Taxonomy Construction (Line 159)**

---

> ### Author Response · Authors · 2025-11-21
> **Response to reviewer vDBt (2/5)**
>
> >W2: The inclusion of 112 aspects also raises concerns about semantic redundancy. Many aspects seem to overlap substantially in meaning, making them difficult to distinguish in practice. For example:
> -Accuracy and Instruction Following are strongly correlated: when a model accurately completes a task, it typically also follows instructions, and deviations from instructions often result in inaccuracy.
> -Creativity-related aspects such as Appeal, Engagingness, and Creativity appear to measure very similar qualities and may not represent truly distinct evaluative dimensions.
> The paper argues that the FRA-OOD test set contains "unseen task-specific aspects" and uses this as evidence of UFEval's generalization ability. However, if these "unseen" aspects are semantically overlapping or near-synonymous with training aspects, the evaluator may not be demonstrating genuine generalization. Instead, it may simply be transferring learned evaluation criteria to synonymous labels, which is conceptually weaker than true out-of-domain generalization. This issue undermines the validity of the claimed aspect-level generalization.
>
> >Q3: Given the high semantic overlap among many aspects, how can the authors prove that the performance on "unseen" aspects represents true generalization to novel concepts, and not just inference on synonymous labels?
>
> A: We thank the reviewer for raising this important point. We would like to provide a clarification regarding the distinction between specific aspects and explain the rationale behind our generalization testing.
>
> **1. Clarification: Accuracy vs. Instruction Following**
>
> We respectfully point out that in fine-grained LLM evaluation, Accuracy and Instruction Following are often treated as distinct:
> - Accuracy measures the factual correctness or truthfulness of the content (e.g., "Is the answer factually right?").
> - Instruction Following measures strict adherence to explicit constraints (e.g., format, word count, forbidden words), independent of truthfulness.
>
> **2. Contextual Generalization (Addressing Semantic Overlap)**
>
> Regarding semantically similar pairs like Appeal (in the OOD set) and Engagingness (in the training set), we acknowledge their high-level semantic similarities. However, this inclusion is intentional. Specifically, we allocated 12 out of the 27 aspects to this category, classifying them as 'Contextual Generalization'. This design serves to explicitly verify our hypothesis proposed in our paper: **"aspects such as engagement, naturalness, and creativity are closely linked. Thus, similar semantics and evaluation standards can be transferred across diverse tasks" (Lines 43–44).**
>
> Our goal was to empirically investigate whether the model can leverage these intrinsic semantic links to adapt to diverse evaluation contexts. We argue that this requires the model to do more than simple label matching; it must adapt the shared semantics to the specific criteria of a new task context.
>
> - Example: "Engagingness" in Stories focuses on plot twists and character development (long-context). "Appeal" in Titles focuses on conciseness and "click-worthiness" (short-context).
>
>
> **3. Generalization to Novel Aspects**
>
> Crucially, to rigorously test generalization beyond semantic transfer, we explicitly included 15 aspects (out of 27) that are semantically distinct from the training set. These aspects introduce unique definition and evaluation criteria never encountered during training.**(A complete list of all aspects from both training and OOD sets is provided in Table 12 of the Appendix, where the 'Specific-task Aspect' column explicitly enumerates the unseen aspects. (Line 1136))**:
>
> - Example: "Difficulty" (in Question Generation) evaluates cognitive complexity and solvability. "Expertise" (in Academic VQA) evaluates the depth of domain-specific knowledge. Neither has a semantic equivalent in the training set.
>
> The table below lists these 15 distinct aspects:
>
> ### Table 1: Novel Unseen Aspects
> | Task | Sub-task | Aspects |
> | :--- | :--- | :--- |
> | **NLG** | Question Generation | Difficulty |
> | | Translation | Translation Fidelity, Thoroughness |
> | | Titile Generation | Suitability |
> | | Keywords Extraction | Prepresentation |
> | **IU** | Academic VQA | Expertise, Explainability |
> | | Medical VQA | Terminology, Transparency |
> | | Embodied Decision | Interpretability, Completion, Riskiness |
> | **ITIG** | Storytelling Gen. | Simplification |
> | | ActivityNet | Reasonableness, Fulfillment |
>
> Finally, we acknowledge that generalization to completely novel aspects is stronger than contextual generalization. To ensure rigor, we have updated the paper to explicitly distinguish between these two categories in Line 414.
>
> **Related Revised Section: 4.2 Out-of-domain Evaluation (Line 414)**

---

> ### Author Response · Authors · 2025-11-21
> **Response to reviewer vDBt (3/5)**
>
> >W3: The paper’s core claim is that jointly learning to assess multiple visual aspects and tasks leads to synergistic effects. To support this, the authors compare models trained on a single task (e.g., “w/ IU”) with the multi-task model ("Ours"), and report performance improvements in Tables 4 and 5. However, the "Ours" model is trained on a larger and more diverse dataset than the other ablations. This makes it difficult to disentangle whether the observed gains actually stem from cross-task synergy or simply from exposure to more varied training data. Without a controlled experiment where the total training data volume is held constant across conditions, the evidence provided does not conclusively support the claimed synergistic effect. A data-controlled ablation would strengthen the argument for true multi-task synergy.
>
> A: We thank the reviewer for raising this important point regarding the experimental design.
>
> **1. Clarification on Original Objective: Cross-Task Synergy**
>
> We would like to clarify that our original ablation study was designed to explore cross-task synergy—specifically, the impact of incorporating distributionally distinct tasks (e.g., adding ITIG and IG data to IU tasks)—rather than to perform a direct comparison of the single-task model and the multi-task model. Although these additional tasks originate from different data sources, they consistently improved performance, providing initial evidence that task diversity is beneficial. **Essentially, our primary objective was to explore whether incorporating data from diverse evaluation task types would positively influence the model's performance on the target evaluation task.**
>
> **2. New Data-Controlled Experiment (Addressing Data Size Concerns)**
>
> However, we fully acknowledge the reviewer's valid concern that performance gains could be confounded by the increase in total data size. To isolate the effect of task diversity from data volume, we conducted a new data-controlled ablation study, in which the total number of training samples was strictly matched across all conditions.
>
> (1). Experimental Setup
>
> To ensure fair comparison and eliminate potential confounding factors from uneven data distribution or volume differences, we adopted a uniform sampling strategy with matched total volume. Specifically, we sampled 10K instances per task from each of the four categories (NLG, IU, IG, and ITIG). Based on these subsets, we constructed the following training sets:
>
> - Single-task baseline: IU (10k); IG (10k)
> - Multi-task combinations (20k total): IU + IG (20k); IU + ITIG (20k); IU + NLG (20k); IG + NLG (20k); IG+ITIG (20k)
>
> All models were fine-tuned on Qwen2-VL-7B using the same training hyperparameters as UFEval to ensure comparability.
>
> (2). Experimental Results
>
> The results for IG benchmarks and IU benchmarks are presented in Table 1 and Table 2, respectively.
>
> ### Table 1: Results on Image Generation (IG) Evaluation Benchmarks
> | Method | GenAI. tau(↑) | GenAI. diff(↑) | Wino. Rel. diff(↑) | Wino. Obj. diff(↑) | Wino. Both diff(↑) | Wino. Ave. diff(↑) | Pick-a-Pic. tau(↑) | Pick-a-Pic. diff(↑) |
> | :--- | :---: | :---: | :---: | :---: | :---: | :---: | :---: | :---: |
> | Qwen2VL-7B| 35.8 | 38.0 | 33.4 | 42.5 | 46.1 | 40.6 | 38.1 | 40.2 |
> | w/ IG (10k)| 42.2 | 49.5 | 43.1 | 46.5 | 59.5 | 49.7 | 41.1 | 46.5 |
> | w/ IG+NLG (20k)  | 42.1 | 49.8 | 43.0 | 46.7 | 59.6 |49.7 | 41.0 | 46.4 |
> | w/ IG+ITIG (20k)  | 42.5 | 52.3 | 45.8 | 47.8 | 61.1 | 51.5 | 42.1 | 48.9 |
> | w/ IG+IU (20k) | **43.4** | **53.9**| **46.2** | **48.0** | **63.4** | **52.5** | **42.9** | **49.6** |
>
> [Abbreviations: GenAI. (GenAI-Bench), Wino. (Winoground), Rel. (Relation), Obj. (Object).]
>
> ### Table 2: Results on Image Understanding (IU) Evaluation Benchmarks
> | Method | WV. tau(↑) | WV. diff(↑) | WV. τ (↑) | MLLM tau(↑) | MLLM diff(↑) | VLR. Gen. diff(↑) | VLR. Hal. diff(↑) | VLR. Rea. diff(↑) | VLR. Ave. diff(↑) |
> | :--- | :---: | :---: | :---: | :---: | :---: | :---: | :---: | :---: | :---: |
> | Qwen2VL-7B | 39.2 | 40.6 | 23.1 | 41.3 | 44.6 | 45.1 |62.8| 62.5 | 56.8 |
> | w/ IU (10k) | 43.5 | 50.2 | 44.5 | 46.5 | 54.5 | 45.3| 60.8 | 65.8 | 57.3 |
> | w/ IU+NLG (20k) | 43.3 | 50.4 | 44.5 | 46.4 | 54.4 | 44.9 | 60.3 |65.7 | 57.2 |
> | w/ IU+ITIG (20k) | 43.9 | 53.1 | 48.1 | 46.8 | 54.9 | 45.5 | 61.0 | 66.0 | 57.5 |
> | w/ IU+IG (20k) | **44.9** | **55.6** | **49.3** | **47.4** | **55.7** | **45.6** |**61.5**| **66.9** | **58.0** |
>
> [Abbreviations: WV. (WildVision), MLLM. (MLLM-as-a-Judge), VLR. (VLRewardBench), Gen. (General), Hal. (Hallucination), Rea. (Reasoning)]
>
> Analysis: The degree of improvement varies across task combinations. Simply doubling the training data with w/ IU+NLG leads to only marginal gains. In strong contrast, w/ IU+IG achieves a substantial improvement (e.g., surging to 55.6).
>
> **Conclusion:** This proves that performance boosts rely on the complementary nature of the auxiliary task, not just increased data volume.

---

> ### Author Response · Authors · 2025-11-21
> **Response to reviewer vDBt (4/5)**
>
> >Q2: What empirical evidence (e.g., expert validation) can be provided to support the robustness and distinctness of the 112 aspects?
>
> A: We appreciate the suggestion to provide empirical evidence supporting the robustness and distinctness. In Appendix H (Line 2742), we reported the Inter-Annotator Agreement (IAA) for aspects grouped by task. Below, we present the IAA at the aspect-level to validate robustness, and provide the aspect correlation analysis to verify distinctness.
>
> **1. Robustness: Inter-Annotator Agreement**
>
> We calculated the IAA for each aspect in the human-annotated data of FRA-ID-H and FRA-OOD-H at the aspect level using Krippendorff’ s Alpha($\alpha$). The detailed score distributions are visualized as boxplots in Figure 15 of the revised manuscript (Line 2762), and the specific values are presented below.
>
> ### Table 1: IAA in FRA-ID-H
> |Aspect|Readability|Spelling Accuracy|Structure Accuracy|Text Repetition|Accuracy|Pointing Out |Difference|Tense Consistency|Creativity|Engagingness|
> |-|-|-|-|-|-|-|-|-|-|-|
> |**Count**|504|145|145|106|266|26|26|26|79|79|
> |**$\alpha$**|0.76|0.95|0.89|0.95|0.99|0.96|0.90|0.98|0.69|0.78|
> |**Aspect**|**Theme**|**Ethnic Bias**|**Nationality Bias**|**Harmfulness**|**Gender Bias**|**Relevance**|**Contextual Information**|**Explanations**|**Information Richness**|**Understandability**|
> |**Count**|39|79|79|199|79|38|38|38|40|40|
> |**$\alpha$**|0.80|1.00|1.00|0.79|1.00|0.75|0.88|0.88|0.78|0.74|
> |**Aspect**|**Count Alignment**|**Spatial Alignment**|**Attribute Alignment**|**Color Alignment**|**Texture Alignment**|**Shape Alignment**|**Action Alignment**| **Object Relationship**|**Fidelity**|**Toxicity**|
> |**Count**|77|80|49|80|79|80|69|19|40|40|
> |**$\alpha$**|0.95|0.92|0.78|1.00|0.77|0.77|0.75|0.73|1.00|1.00|
> |**Aspect**|**Coverage**|**Length Constraint**|**Conciseness**|**Clarity**|**Grammaticality**|**Coherence**|**Fluency**| **Sentence Complexity**|**Vocabulary Complexity**|
> |**Count**|41|41|81|384|386|504|504|278|318|
> |**$\alpha$**|0.73|1.00|0.78|0.76|0.89|0.72|0.72|0.81|0.87|
> | **Aspect**|**Helpfulness**|**Instruction Following**|**Instruction Inconsistency**|**Context Inconsistency**|**Privacy Violations**|**Bias**|**Object Alignment**|**Text-Image Relationship**|**Alignment**| |
> |**Count**|80|200|40|40|40|200|85|40|80| |
> |**$\alpha$**|0.79|0.90|1.00|1.00|1.00|0.95|0.95|0.76|0.76| |
> |**Aspect**|**Stylistic Consistency**|**Character Consistency**|**Action Consistency**|**Scene Consistency**|**Image Repetition**|**Semantic Consistency**|**Feasibility**|**Safety**|**Completeness**| |
> |**Count**|80|40|80|80|80|80|40|40|120| |
> |**$\alpha$**|0.89|0.90|0.90|0.93|0.96|0.78|0.74|1.00|0.77| |
>
> ### Table 2: IAA in FRA-OOD-H
> |Aspect|Cultural Bias|Translation Fidelity|Thoroughness|Tense Consistency|Spelling Accuracy| Structure Accuracy|Precision|Format|Privacy Violations|Simplification|
> |-|-|-|-|-|-|-|-|-|-|-|
> |**Count**|37|39|39|39|39|39|40|40|40|40|
> |**$\alpha$**|1.00|0.73|0.73|0.85|1.00|0.90|0.90|0.86|1.00|0.68|
> |**Aspect**|**Vocabulary Complexity**|**Relevance**|**Representation**|**Expertise**| **Correctness**|**Explainability**|**Riskiness**|**Completion**|**Interpretability**| **Harmfulness**|
> |**Count**|120|38|38|40|40|40|40|40|40|347|
> |**$\alpha$**|0.79|0.66|0.69|0.75|1.00|0.80|0.82|0.72|0.71|0.71|
> |**Aspect**|**Regional Bias**|**Gender Bias**|**Toxicity**|**Terminology**|**Transparency**|**Color Alignment**|**Shape Alignment**|**Texture Alignment**|**Attribute Alignment**|**Scene Alignment**|
> |**Count**|40|40|40|40|40|161|161|161|161|112|
> |**$\alpha$**|1.00|1.00|1.00|0.80|0.78|0.86|0.75|0.70|0.74|0.89|
> |**Aspect**|**Action Alignmen** |**Object Relationship**|**Fidelity**|**Alignment**|**Spatial Alignment**| **Count Alignment**|**Object Alignment**|**Comparison**|**Negation**|**Universal**|
> |**Count**|83|143|227|227|98|54|54|40|41|40|
> |**$\alpha$**|0.88|0.82|0.95|0.67|0.75|0.89|0.88|0.73|0.78|0.73|
> |**Aspect**|**Fulfillment**|**Reasonableness**|**Character Consistency**|**Image Repetition**|**Scene Consistency**|**Action Consistency**|**Stylistic Consistency**| **Semantic Consistency**|**Image Coherence**|**Originality**|
> |**Count**|40|40|80|80|80|80|79|79|40|40|
> |**$\alpha$**|0.76|0.72|0.81|0.80|0.83|0.85|0.72|0.76|0.96|0.65|
> |**Aspect**|**Readability**|**Fluency**|**Coherence**|**Grammaticality**|**Coverage**| **Appeal**|**Suitability**|**Clarity**|**Bias/Ethnic Bias**|**Sentence Complexity**|
> |**Count**|315|274|354|234|37|37|37|274|77/37|120|
> |**$\alpha$**|0.71|0.68|0.73|0.80|0.65|0.60|0.75|0.64|1.00/1.00|0.83|
>
> Analysis:
> - FRA-ID-H: The ID dataset demonstrates strong agreement with a mean $\alpha$ of 0.87. All aspects scored above 0.69.
> - FRA-OOD-H: The OOD dataset maintains high consistency with a mean $\alpha$ of 0.81. All aspects scored above 0.6.
>
> **Conclusion:** These results indicate high consistency in our annotations, with the vast majority of aspects achieving substantial agreement in both in-domain and out-of-domain settings.

---

> ### Author Response · Authors · 2025-11-21
> **Response to reviewer vDBt (5/5)**
>
> >Q2: What empirical evidence (e.g., expert validation) can be provided to support the robustness and distinctness of the 112 aspects?
>
>
> **2. Distinctness: Correlation Analysis**
>
> To further quantify the distinctness, we computed Kendall’s Tau ($\tau$) correlations between aspects within each task. Correlation heatmaps are provided in the revised manuscript (Line 2884), with statistical summaries shown in Table 1 (FRA-ID-H for in-domain) and Table 2 (FRA-OOD-H for out-of-domain) below.
>
> ###  Table 1: FRA-ID-H Aspect Colleration
> | Task Category | Task | N_Asp | Mean τ | Max τ |
> |---|---|---|---|---|
> | IU | Chart Reasoning | 2 | 0.609 | 0.609 |
> | | Detailed Captioning | 18 | 0.164 | 0.597 |
> | | General VQA | 8 | 0.291 | 0.682 |
> | | Medical Image Und. | 8 | 0.131 | 0.656 |
> | | Robustness Inst. | 9 | 0.306 | 0.678 |
> | | Simple Captioning | 12 | 0.310 | 0.642 |
> | | Text Rich Und. | 9 | 0.380 | 0.640 |
> | ITIG | Vist | 15 | 0.250 | 0.650 |
> | | Wikihow | 14 | 0.406 | 0.692 |
> | NLG | Creative Writing | 15 | 0.257 | 0.678 |
> | | Functional Writing | 13 | 0.503 | 0.703 |
> | | General Comm. | 14 | 0.328 | 0.800 |
> | | Rewriting | 12 | 0.332 | 0.647 |
> | | Summarization | 14 | 0.325 | 0.627 |
> | IG | Text To Image | 12 | 0.576 | 0.656 |
>
> ###  Table 2: FRA-OOD-H Aspect Colleration
> | Task Category | Subtask | N_Asp | Mean τ | Max τ |
> |---|---|---|---|---|
> | IG | Text-to-Image | 16 | 0.352 | 0.723 |
> | IU | M3it | 9 | 0.298 | 0.566 |
> | | Pcaeval | 12 | 0.355 | 0.593 |
> | | Pmc | 9 | 0.518 | 0.699 |
> | ITIG | Activitynet Cont. | 13 | 0.370 | 0.694 |
> | | Storytelling Gen. | 13 | 0.404 | 0.685 |
> | NLG | Data Analysis | 9 | 0.256 | 0.581 |
> | | Keywords Ext. | 7 | 0.397 | 0.649 |
> | | Question Gen. | 9 | 0.462 | 0.681 |
> | | Title Gen. | 11 | 0.240 | 0.597 |
> | | Translation | 9 | 0.420 | 0.618 |
>
> Analysis:
>
> - FRA-ID-H: Mean correlations across tasks range from 0.131 to 0.609. All values remain well below the redundancy threshold (>0.8).
> - FRA-OOD-H: Mean correlations range from 0.240 (Title Generation) to 0.518 (Pmc). Most tasks maintain low-to-moderate correlation values.
>
> **Conclusion:** Based on the above analysis, aspect correlations are consistently low across both datasets, supporting the distinctness of our taxonomy.
>
> **Related Revised Section: Appendix E (Line 2754)**

---

> > ### Comment · Reviewer_vDBt · 2025-11-24
> >
> > Thank you very much for the detailed response. I am satisfied with the rebuttal, and all my concerns have been fully addressed. I have raised my rating to 8. Good luck.

---

> ### Author Response · Authors · 2025-11-24
> **Thank you for the positive feedback and raised rating**
>
> Dear Reviewer vDBt,
>
> Thank you very much for your encouraging response and for raising the rating to 8! We are thrilled that our clarifications and additional experiments have fully resolved your concerns.
>
> We truly value the detailed feedback you provided, which significantly guided us to improve the robustness and clarity of our work.
>
> Best regards,
>
> The Authors

---

### Official Review · Reviewer_B3KA · 2025-10-27

**Soundness:** 3
**Presentation:** 3
**Contribution:** 3
**Rating:** 4
**Confidence:** 3

**Summary:**

This paper addresses the critical challenge of evaluating open-ended outputs from MLLMs. The authors argue that existing "MLLM-as-a-Judge" evaluators are too narrowly focused on specific tasks and aspects, limiting their generalizability. To overcome this, they propose two main contributions: 1) FRABench, a large-scale, fine-grained evaluation benchmark covering four major multimodal tasks (NLG, IU, IG, ITIG) across 28 sub-tasks, and 2) UFEval, a unified evaluator model trained on FRABench. The core hypotheses are that learning to evaluate diverse tasks and aspects fosters synergistic benefits, and that evaluators can generalize to unseen tasks and aspects due to their inherent semantic connections. The paper presents a comprehensive hierarchical taxonomy of 112 evaluation aspects, which forms the foundation of FRABench's 60.4k pairwise samples. Experiments demonstrate that UFEval achieves strong performance on out-of-domain tasks and aspects, is competitive with specialized evaluators on public benchmarks, and can be used to generate high-quality preference data for downstream model alignment via DPO.

**Strengths:**

1. The paper tackles a well-recognized bottleneck in MLLM research. As model capabilities expand, the need for scalable, reliable, and nuanced evaluation frameworks is paramount. The authors correctly identify the limitations of current approaches and propose a compelling, unified vision for evaluation.
2. The development of FRABench is a major contribution in its own right. Creating a benchmark of this scale (60.4k pairs, 325k labels) that is multi-task, multi-modal, and fine-grained is a significant undertaking. The hierarchical aspect taxonomy is also a valuable conceptual contribution, providing a structured and comprehensive framework for thinking about MLLM quality that could be widely adopted.
3. The paper presents a strong set of results. UFEval consistently outperforms strong baselines on out-of-domain generalization, particularly on the human-annotated sets. Furthermore, it achieves competitive, and sometimes superior, performance against state-of-the-art, specialized evaluators (e.g., Themis, LLaVA-Critic) on established public benchmarks, despite being a single, unified model.

**Weaknesses:**

1. This is the most significant concern regarding the methodology. A substantial portion of the 325k training labels are generated by GPT-4o. While the authors use human-annotated data for testing, the training data's quality is fundamental to the final model's performance. The paper's validity rests on the assumption that GPT-4o is a sufficiently reliable and unbiased annotator for 112 diverse aspects.
2. While the resulting aspect tree is a strength, the process of its creation is described somewhat briefly. The paper mentions adapting existing structures and manually organizing the rest based on definitions. This process can be highly subjective.
3. The paper excels at presenting strong aggregate performance metrics. However, a deeper understanding would be gained from analyzing where UFEval fails. A qualitative analysis comparing UFEval's judgments to human judgments, especially in cases of disagreement, would be highly insightful.
4. There are few baseline comparisons, and qwen2-vl is already a year-old model. Authors should compare more recent large multimodal models and experiment with newer models

**Questions:**

1. In Table 1, the number of aspects for each evaluator is listed. For instance, AUTO-J is listed with 332. Are these aspects comparable in scope and granularity to the 112 aspects in your taxonomy? A direct numerical comparison might be misleading if the definition of an "aspect" differs significantly across studies. A clarifying footnote could be helpful.
2. The base model for UFEval is Qwen2-VL-7B. How much of the observed performance gain is attributable to the FRABench data versus the inherent strengths of this particular base model? A useful baseline might be to fine-tune Qwen2-VL-7B on a single-task, single-aspect dataset (e.g., the data used for Themis) and compare its performance on that specific task against UFEval. This would help further isolate the benefits of the unified training approach.
3. Regarding the GPT-4o annotations for UAs , you mention providing only the response to avoid bias from query-correctness. This is a very thoughtful design choice. Did you explore or observe any other biases in the GPT-4o annotations, and did you employ any other strategies to mitigate them during data creation?

---

> ### Author Response · Authors · 2025-11-21
> **Response to reviewer B3KA (1/4)**
>
> Dear Reviewer B3KA,
>
> We appreciate your time and constructive feedback, which has been invaluable in improving our work. We have taken your concerns very seriously and have made every effort to address your questions and incorporate your suggestions.
>
> >Weakness 1: This is the most significant concern regarding the methodology. A substantial portion of the 325k training labels are generated by GPT-4o. While the authors use human-annotated data for testing, the training data's quality is fundamental to the final model's performance. The paper's validity rests on the assumption that GPT-4o is a sufficiently reliable and unbiased annotator for 112 diverse aspects.
>
> >Question 3: Regarding the GPT-4o annotations for UAs, you mention providing only the response to avoid bias from query-correctness. This is a very thoughtful design choice. Did you explore or observe any other biases in the GPT-4o annotations, and did you employ any other strategies to mitigate them during data creation?
>
> A: We sincerely appreciate the reviewer for raising this critical concern regarding the validity of GPT-4o annotations and the potential biases involved. Below, we clarify our rationale for selecting GPT-4o and detail the quality control measures we implemented.
>
> **1. Why we chose GPT-4o?**
>
> Through our preliminary investigation, we found that many recent studies [1-3] utilize GPT-4o for data annotation. This preference is driven by findings that GPT-4o achieves superior consistency with human annotators. Moreover, for a fine-grained evaluation involving 112 aspects, manual annotation is prohibitively expensive and susceptible to fatigue errors. Therefore, we adopted GPT-4o for our annotation process.
>
> **2. Quality Control and Bias Mitigation**
>
> We acknowledge the potential issues introduced by GPT-4o. Therefore, we implemented quality control measures during the annotation process.
>
> In addition to the strategy you mentioned—providing only the response to mitigate bias towards query-correctness—we also observed a phenomenon where the model fails to adhere to the strict boundaries of a fine-grained aspect.
> - Observation: During the early prompt design phase, we observed that GPT-4o tended to "expand the scope" of its evaluation. When evaluating a specific "child node" aspect, GPT-4o sometimes included criteria from its "parent node".
> - Example: For 'Shape Alignment', instead of focusing strictly on whether the object's shape matched the text, the model would sometimes critique higher-level parent aspects like 'Material Alignment', which encompasses additional dimensions such as color and texture alignment.
> - Solution (Negative Constraints): We refined our prompts by adding strict negative constraints, explicitly instructing the model to ignore everything except the target criterion.
> - Refined Prompt (The complete prompt can be found in Appendix G): "Do not evaluate them in general terms or based on factors unrelated to the given criterion. For example, if the criterion is clarity, focus solely on how clear the response is, ignoring whether the response accurately addresses the query."
>
> Finally, we validated these refined prompts on our human-annotated datasets (FRA-OOD-H). We achieved high agreement rates with human annotators of 83% (detailed agreement rates are presented in Table 2 of the main text (Line 293)), thereby ensuring the quality of our data annotation.
>
> [1] Llava-critic: Learning to evaluate multimodal models. (CVPR)
>
> [2] Prometheus-Vision: Vision-Language Model as a Judge for Fine-Grained Evaluation (ACL)
>
> [3] Halu-J: Critique-Based Hallucination Judge. (AAAI)

---

> ### Author Response · Authors · 2025-11-21
> **Response to reviewer B3KA (2/4)**
>
> >Weakness 2: While the resulting aspect tree is a strength, the process of its creation is described somewhat briefly. The paper mentions adapting existing structures and manually organizing the rest based on definitions. This process can be highly subjective.
>
> A: Thank you for your comment regarding “the process of its creation is described somewhat briefly”. We acknowledge that we did not provide sufficient information regarding our methodology, which may have led to ambiguity. To address this, we have added a detailed explanation in the revised manuscript (Line 159) under the section **Aspect Tree Construction**.
>
> **Aspect Tree Construction**
>
> 1. Adoption of Established Taxonomies
>
>  To ensure consistency and robustness, we prioritized aspect trees from peer-reviewed literature. Specifically, we adopted the following structures for Readability [1], Bias [2], Instruction Following [3], Hallucination [4], Alignment [5], Complement [6], Image Coherence [7], Image Quality [8], Semantic Consistency [9], and Toxicity [10]. Relying on these published works ensures the validity and rationality of the foundational structure.
>
> 2. Integration of Additional Aspects
>
> For the remaining aspects without an established hierarchical structure, we handle them as follows:
>
> - Bidirectional Matching Strategy: First, we check whether the remaining aspect's name appears within the definition of the existing tree's root node. If a match is found, we recursively traverse the child nodes to identify the most specific insertion point and append the aspect as a child of the last matched node. Conversely, if no downward match is found, we verify whether the root node's name appears within the definition of the aspect. In this case, the aspect implies a broader concept and is inserted as the parent of the current root. This applies to the following aspects: [Clarity in Readability], [Semantic Consistency in Image Coherence], and [Integration, Complement, and Alignment in the Text-Image Relationship].
> - Creation of new root nodes: For aspects that do not match any existing tree, we established them as independent root nodes to minimize subjective bias and avoid misleading categorizations. These nodes correspond to all green peer-level nodes within the TAs tree in Figure 1 (Line 162).
>
> **Regarding Subjectivity:** We maximize objectivity by adhering to established academic standards, anchoring our core hierarchy in validated structures [1-10]. Even when handling aspects lacking explicit hierarchical definitions, we minimized manual intervention and subjective inference by strictly following the two-step processing logic detailed above.
>
> [1] Are llm-based evaluators confusing nlg quality criteria? (ACL)
>
> [2] Measuring bias in ai models: an statistical approach introducing n-sigma. (COMPSAC)
>
> [3] Evaluating large language models at evaluating instruction following. (ICLR)
>
> [4] A survey on hallucination in large language models: Principles, taxonomy, challenges, and open questions. (TOIS)
>
> [5] T2i-compbench: A comprehensive benchmark for open-world compositional text-to-image generation. (NIPS)
>
> [6] Categorizing and Inferring the Relationship between the Text and Image of Twitter Posts. (ACL)
>
> [7] Holistic Evaluation for Interleaved Text-and-Image Generation. (ACL)
>
> [8] Q-Eval-100K: Evaluating Visual Quality and Alignment Level for Text-to-Vision Content (CVPR)
>
> [9]Neural Storyboard Artist: Visualizing Stories with Coherent Image Sequence. (MM)
>
> [10] RealToxicityPrompts: Evaluating Neural Toxic Degeneration in Language Models. (EMNLP)
>
> **Related Revised Section: 3.1.2 Aspect Taxonomy Construction (Line 159)**

---

> ### Author Response · Authors · 2025-11-21
> **Response to reviewer B3KA (3/4)**
>
> >W3: The paper excels at presenting strong aggregate performance metrics. However, a deeper understanding would be gained from analyzing where UFEval fails. A qualitative analysis comparing UFEval's judgments to human judgments, especially in cases of disagreement, would be highly insightful.
>
> A: We sincerely thank the reviewer for this insightful suggestion. We agree that a qualitative analysis of failure cases is essential for a deeper understanding of the model's limitations.
>
> Due to space constraints in the initial submission, we placed the detailed bad case analysis, which investigates relatively underperforming aspects and illustrates specific bad cases
> — in the Appendix (Line 3672).
>
> Following your advice to improve the paper's accessibility, we have moved part of the failure cases from the appendix into the main text (Line 426) to provide a more comprehensive discussion.
>
> **Related Revised Section: 4.2 Out-of-domain Evaluation (Line 426)**
>
> >W4: There are few baseline comparisons, and qwen2-vl is already a year-old model. Authors should compare more recent large multimodal models and experiment with newer models.
>
> A: We appreciate the reviewer’s suggestion to include more recent comparisons.
>
> - For the NLG and IU tasks: We selected the best available specialized open-source models: **Themis for NLG, which is the current SOTA in text generation evaluation**, and **LLaVA-Critic for IU, which is the latest specialized model for image understanding evaluation.**
> - For the IG task: We additionally incorporated two of the most recent and representative approaches: **Q-Eval** [1] (trained on the latest human preference datasets) and **CIGEval** [2] (an agent-based evaluator).
> - Across all tasks, we also evaluated Qwen3-VL-8B (Q3VL) as a general-purpose model to provide a broader performance comparison.
>
> ### Table 1: Evaluation as MLLM-as-a-Judge for NLG
> | Method | Sum. Coh. tau(↑) | Sum. Coh. diff(↑) | Sum. Cons. tau(↑) | Sum. Cons. diff(↑) | Sum. Flu. tau(↑) | Sum. Flu. diff(↑) | Sum. Rel. tau(↑) | Sum. Rel. diff(↑) | Sum. Ave. tau(↑) | Sum. Ave. diff(↑) | MANS. diff(↑) | MT-Bench. tau(↑) | MT-Bench. diff(↑) |
> |---|---|---|---|---|---|---|---|---|---|---|---|---|---|
> |Q3VL|56.5|61.5|60.8|57.4|68.0|75.5|55.4|60.8|60.1|63.8|45.5|70.1|80.9|
> | Ours| **64.6** | **71.8** | **75.2** | **83.5** | **74.7** | **84.5** | **61.3** | **67.6** | **69.0** | **76.9** | **69.3** | **74.9** | **88.3** |
>
> [Abbreviations: Sum. (SummEval), Coh. (Coherence), Cons. (Consistency), Flu. (Fluency), Rel. (Relevance)]
>
> ### Table 2: Evaluation as MLLM-as-a-Judge for IU
> | Method | WV. tau(↑) | WV. diff(↑) | WV.  τ (↑) | MLLM tau(↑) | MLLM diff(↑) | VLR. Gen. diff(↑) | VLR. Hal. diff(↑) | VLR. Rea. diff(↑) | VLR. Ave. diff(↑) |
> |-|-|-|-|-|-|-|-|-|-|
> | Q3VL| 48.1 | 57.7 | 54.5 | 53.0 | 57.5 | 45.5 | **66.0** | 69.0 | **60.1** |
> | Ours | **53.9** | **68.6** | **66.5** | **57.2** | **67.0** | **46.4** | 57.7 | **71.1** | 58.4 |
>
> [Abbreviations: WV. (WildVision), MLLM. (MLLM-as-a-Judge), VLR. (VLRewardBench), Gen. (General), Hal. (Hallucination), Rea. (Reasoning)]
>
> ### Table 3: Evaluation as MLLM-as-a-Judge for IG
> | Method | GenAI. tau(↑) | GenAI. diff(↑) | Wino. Rel. diff(↑) | Wino. Obj. diff(↑) | Wino. Both diff(↑) | Wino. Ave. diff(↑) | Pick-a-Pic. tau(↑) | Pick-a-Pic. diff(↑) |
> |--|--|--|--|-| :---: | :---: | :---: | :---: |
> | Q-Eval | 48.0 | 65.5 | **62.4** | 54.5 | 61.5 | 59.4 | **55.0** | **62.2** |
> | CIGEval | 40.1 | 29.8 | 42.5 | 30.4 | 30.7 | 34.5 | 34.5 | 31.5 |
> | Q3VL | 53.0 | 59.0 | 54.6 | 63.6 | 69.0 | 62.4 | 49.2 | 55.1 |
> | Ours | **53.6** | **65.5** | 57.5 | **59.1** | **80.7** | **65.7** | 50.0 | 57.3 |
>
> [Abbreviations: GenAI. (GenAI-Bench), Wino. (Winoground), Rel. (Relation), Obj. (Object)]
>
> **Conclusion:** Our model demonstrates competitive performance compared to most specialized open-source models on the IG task and surpasses the latest Qwen3-VL-8B.
>
> Regarding the reviewer’s suggestion to train on the latest large multimodal models, we fully agree that utilizing the latest backbone models would likely yield further performance gains. In our future work, we plan to extend UFEval by training on the upcoming Qwen3-VL series to continue pushing the state-of-the-art.
>
> [1] Q-Eval-100K: Evaluating Visual Quality and Alignment Level for Text-to-Vision Content (CVPR 2025)
>
> [2] A Unified Agentic Framework for Evaluating Conditional Image Generation. (ACL 2025)
>
> **Related Revised Section: 4.3 Evaluation as MLLM-as-a-Judge (Table 4 and Table 5 in Line 378)**

---

> ### Author Response · Authors · 2025-11-21
> **Response to reviewer B3KA (4/4)**
>
> >Q1: In Table 1, the number of aspects for each evaluator is listed. For instance, AUTO-J is listed with 332. Are these aspects comparable in scope and granularity to the 112 aspects in your taxonomy? A direct numerical comparison might be misleading if the definition of an "aspect" differs significantly across studies. A clarifying footnote could be helpful.
>
> A: We thank the reviewer for this helpful suggestion regarding the comparability of aspect counts in Table 1.
>
> The aspects listed for other evaluators are comparable to our aspects. To prevent any potential confusion, we have added a clarifying note in the caption of Table 1 in the revised manuscript (Line 56), as suggested, to explicitly explain this relationship.
>
> **Related Revised Section: 1 Introduction (Table 1 in Line 56)**
>
> >Q2: The base model for UFEval is Qwen2-VL-7B. How much of the observed performance gain is attributable to the FRABench data versus the inherent strengths of this particular base model? A useful baseline might be to fine-tune Qwen2-VL-7B on a single-task, single-aspect dataset (e.g., the data used for Themis) and compare its performance on that specific task against UFEval. This would help further isolate the benefits of the unified training approach.
>
> A: We sincerely thank the reviewer for this insightful suggestion, which helps to further isolate the specific benefits of our unified training approach.
>
> To address this point, we conducted the ablation study that the reviewer proposed. We fine-tuned the Qwen2-VL-7B using the 60k training data provided by Themis. Critically, we used the identical training parameters and configuration as our UFEval to ensure a fair comparison.
>
> We then evaluated this model (referred to as w/Themis) on NLG benchmarks, including SummEval, MANS, and MT-Bench. To further verify the value of our dataset, we also report results for a model trained exclusively on the NLG-related portion of FRABench (20k samples), referred to as w/F-NLG.
>
> The results are as follows:
>
> ### Table 1: Evaluation as MLLM-as-a-Judge for NLG
> | Method | Sum. Coh. tau(↑) | Sum. Coh. diff(↑) | Sum. Cons. tau(↑) | Sum. Cons. diff(↑) | Sum. Flu. tau(↑) | Sum. Flu. diff(↑) | Sum. Rel. tau(↑) | Sum. Rel. diff(↑) | Sum. Ave. tau(↑) | Sum. Ave. diff(↑) | MANS. diff(↑) | MT-Bench. tau(↑) | MT-Bench. diff(↑) |
> | :--- | :---: | :---: | :---: | :---: | :---: | :---: | :---: | :---: | :---: | :---: | :---: | :---: | :---: |
> | Q2VL-7B |48.8 | 52.5 | 35.5 | 30.6 | 44.8 | 49.8 | 41.7 | 44.5 | 42.7 | 44.3 | 60.2 | 44.5 | 50.1 |
> | w/Themis | 56.4 | 57.1 | **81.4** | **83.5** | 69.8 | 72.0 |  52.1 | 52.7 | 64.9 | 66.4 | 44.0 | 45.5 | 58.4 |
> | w/F-NLG | 60.3 | 66.8  | 72.9 |  81.8  |  73.0  |  82.5 |  60.0 |  66.5  |  65.7 | 74.4 |68.5 | 73.0 | 86.4 |
> | UFEval| **64.6** | **71.8** | 75.2 | 83.5 | **74.7** | **84.5** | **61.3** | **67.6** | **69.0** | **76.9** | **69.3** | **74.9** | **88.3** |
>
> [Abbreviations: Sum. (SummEval), Coh. (Coherence), Cons. (Consistency), Flu. (Fluency), Rel. (Relevance)]
>
> **Conclusion:** Our experiments show that w/F-NLG (trained on 20k NLG data from FRABench) already outperforms w/Themis (trained on 60k Themis data) on most benchmarks, while UFEval achieves the best performance. This demonstrates the effectiveness of our dataset.

---

> > ### Comment · Reviewer_B3KA · 2025-11-25
> >
> > Thanks for your detailed rebuttal. Most of my concerns are addressed, so I'd like to raise the score to 6.

---

> > > ### Author Response · Authors · 2025-11-26
> > > **Thank you for the positive feedback and raised rating**
> > >
> > > Dear Reviewer B3KA,
> > >
> > > We sincerely thank you for your positive feedback and for raising the rating to 6. We are glad to hear that our responses have addressed your concerns.
> > >
> > > We are very grateful for your time and constructive comments throughout the review process, which have significantly improved the quality of our work.
> > >
> > > Best regards,
> > >
> > > The Authors

---

### Official Review · Reviewer_CEqP · 2025-10-30

**Soundness:** 3
**Presentation:** 1
**Contribution:** 3
**Rating:** 6
**Confidence:** 3

**Summary:**

This paper introduced a benchmark / evaluation dataset called FRABench for measuring so-called "aspects" for text, image generation and image understanding. Such aspects can be coherence, grammaticality, semantic consistency, etc. Also, it introduces UFEval, an multimodal LLM as a judge for such tasks and aspects, built by fine-tuning Qwen2-VL-7B-Instruct on the new FRABench dataset.

**Strengths:**

* FRABench is a substantial contribution: 60.4k pairwise samples and 325k aspect-level labels covering 112 distinct aspects organized in a clear hierarchical taxonomy.

**Weaknesses:**

* Vague terminology and presentation: The paper repeatedly uses the term “aspects” in the abstract and introduction without defining it or providing concrete examples. This makes it difficult for the reader to understand what exactly is being evaluated until much later (around page 4). Introducing a brief definition or illustrative examples early on would substantially improve readability and accessibility.

* Although the dataset includes some human labels, most of the 325k aspect-level annotations are GPT-4o-generated which raises the question of bias dna weakness propagation.

* UFEval is built only on Qwen2-VL-7B-Instruct, which raises concerns about how much the findings depend on that particular architecture. How do the findings generalise for other backbones, such as LlaVA, mplug-owl-3, etc?

* it would have been useful to have a qualitative analysis of failure modes.

* Generalisation tests: While UFEval is evaluated on several external benchmarks (e.g., Winoground, Pick-a-Pic), all of its training and main “aspect generalization” analyses are conducted within the FRABench framework. Since both training and out-of-domain splits rely on the same hierarchical taxonomy of aspects, it remains somewhat unclear how well UFEval’s claimed aspect-level generalization would hold under a completely independent fine-grained evaluation scheme.

**Questions:**

L016-017 there is so much repetitions of the word "aspects"

---

> ### Author Response · Authors · 2025-11-21
> **Response to reviewer CEqP (1/2)**
>
> Dear Reviewer CEqP,
>
> We appreciate your time and constructive feedback, which has been invaluable in improving our work. We also express our sincere gratitude for your positive assessment of our paper and for recognizing the value of our contributions.
>
> >W1: Vague terminology and presentation: The paper repeatedly uses the term “aspects” in the abstract and introduction without defining it or providing concrete examples. This makes it difficult for the reader to understand what exactly is being evaluated until much later (around page 4). Introducing a brief definition or illustrative examples early on would substantially improve readability and accessibility.
>
> >Q1: L016-017 there is so much repetitions of the word "aspects"
>
> A: We thank the reviewer for the helpful suggestion. We have revised the manuscript to include a definition of aspects in the Abstract (Line 16). Additionally, we have incorporated synonyms such as "criteria" to reduce the repetitive usage of the term aspects.
>
> **Related Revised Section: Abstract (Line 16)**
>
> >W2: Although the dataset includes some human labels, most of the 325k aspect-level annotations are GPT-4o-generated which raises the question of bias and weakness propagation.
>
> A: We share the reviewer's concern regarding the potential for bias and weakness propagation when relying on synthetic annotations. Therefore, when using GPT-4o for annotation, we implemented two strategies to control quality and mitigate these risks.
>
> **Strategy I: Mitigating Query-Correctness Bias**
>
> As mentioned in Line 249 in the main text, we observed that GPT-4o frequently incorporates correctness into its assessment of Universal Aspects, even when the aspect is purely stylistic.
>
> - Example: When annotating Text Readability, GPT-4o would often penalize a response if the content was incorrect relative to the query, even if the text itself was highly readable and fluent.
> - Solution: To address this, we provided only the response (excluding the query) to the model when evaluating UAs. This forces the model to evaluate the aspect (e.g., Readability) in isolation, preventing bias from the query-response relationship.
>
> **Strategy II: Enforcing Strict Evaluation Boundaries**
>
> We also observed a phenomenon where the model fails to adhere to the strict boundaries of a fine-grained aspect.
>
> - Observation: When evaluating a specific "child node" aspect, GPT-4o sometimes included criteria from its "parent node".
> - Example: For 'Shape Alignment', instead of focusing strictly on whether the object's shape matched the text, the model would sometimes critique higher-level parent aspects like 'Material Alignment', which encompasses additional dimensions such as color and texture.
> - Solution (Negative Constraints): We refined our prompts by adding strict negative constraints, explicitly instructing the model to ignore everything except the target criterion.
> - Refined Prompt (The complete prompt can be found in Appendix G (Line 3138)): "Do not evaluate them in general terms or based on factors unrelated to the given criterion. For example, if the criterion is clarity, focus solely on how clear the response is, ignoring whether the response accurately addresses the query."
>
> Finally, we validated these refined prompts on our human-annotated datasets (FRA-OOD-H). We achieved high agreement rates with human annotators of 83% (detailed agreement rates are presented in Table 2 of the main text (Line 293)), thereby ensuring the quality of our data annotation.
>
> >W3: it would have been useful to have a qualitative analysis of failure modes.
>
> A: We sincerely thank the reviewer for this insightful suggestion. We agree that a qualitative analysis of failure cases is essential for a deeper understanding of the model's limitations.
>
> Due to space constraints in the initial submission, we placed the detailed bad case analysis—which investigates relatively underperforming aspects and illustrates specific bad cases— in the Appendix (Line 3672).
>
> Following your advice to improve the paper's accessibility, we have moved part of the failure cases from the appendix into the main text (Line 426) to provide a more comprehensive discussion.
>
> **Related Revised Section: 4.2 Out-of-domain Evaluation (Line 426)**

---

> ### Author Response · Authors · 2025-11-21
> **Response to reviewer CEqP (2/2)**
>
> >W4: UFEval is built only on Qwen2-VL-7B-Instruct, which raises concerns about how much the findings depend on that particular architecture. How do the findings generalise for other backbones, such as LlaVA, mplug-owl-3, etc?
>
> A: We very thank the reviewer for raising this interesting point regarding architectural generalization. We agree that verifying the method across different backbones is valuable.
>
> While we are genuinely interested in exploring this direction, conducting full-scale training on multiple additional backbones is computationally intensive and extends beyond the current revision timeline. Therefore, we may not provide empirical evidence at this stage.
>
> We acknowledge that architecture sensitivity is a well-documented phenomenon within the field of LLMs. Methods effective on one model series may not necessarily transfer to others. Consequently, in the absence of corresponding experiments, we refrain from making definitive claims regarding the universality of our findings to other architectures.
>
> **Consequently, to ensure rigor, we have explicitly stated in the Limitations section (Line 536) of the revised manuscript that our findings have been validated exclusively on the Qwen series, and have not yet been verified on other architectures.**
>
> **Related Revised Section: Conclusion, Limitation, and Future Work (Line 536)**
>
> >W5: Generalisation tests: While UFEval is evaluated on several external benchmarks (e.g., Winoground, Pick-a-Pic), all of its training and main “aspect generalization” analyses are conducted within the FRABench framework. Since both training and out-of-domain splits rely on the same hierarchical taxonomy of aspects, it remains somewhat unclear how well UFEval’s claimed aspect-level generalization would hold under a completely independent fine-grained evaluation scheme.
>
> A: We appreciate the reviewer's concern regarding the validation of our aspect-level generalization under independent evaluation schemes.
>
> To rigorously address this, after a comprehensive investigation of existing evaluation schemes, we identified the Themis evaluation scheme as strictly non-overlapping with our FRABench. Consequently, we adopted the evaluation criteria defined by Themis to conduct evaluations on established public benchmarks: SummEval, MANS, and MT-Bench.
>
> **Experimental Results:** We compared UFEval prompted with Themis criteria (marked as UFEvalT) against the original Themis and the UFEval (using our scheme)
>
> ### Table 1: Evaluation as MLLM-as-a-Judge for NLG
> | Method | Sum. Coh. tau(↑) | Sum. Coh. diff(↑) | Sum. Cons. tau(↑) | Sum. Cons. diff(↑) | Sum. Flu. tau(↑) | Sum. Flu. diff(↑) | Sum. Rel. tau(↑) | Sum. Rel. diff(↑) | Sum. Ave. tau(↑) | Sum. Ave. diff(↑) | MANS. diff(↑) | MT-Bench. tau(↑) | MT-Bench. diff(↑) |
> | :--- | :---: | :---: | :---: | :---: | :---: | :---: | :---: | :---: | :---: | :---: | :---: | :---: | :---: |
> | Themis | 60.7| 62.1 | **81.8** | **86.0** | 73.3 | 77.7 | 54.4 | 54.6 | 67.5 | 70.1 | 44.2 | 43.6 | 37.7 |
> | UFEvalT | 64.4 | 70.8 |73.3 | 83.4| 73.3 | 83.1 | **65.7** | **73.6** | **69.6** | **78.4** | **71.0** | 72.3 | 84.5 |
> | UFEval| **64.6** | **71.8** | 75.2 | 83.5 | **74.7** | **84.5** | 61.3 | 67.6 | 69.0 | 76.9 | 69.3 | **74.9** | **88.3** |
>
> [Abbreviations: Sum. (SummEval), Coh. (Coherence), Cons. (Consistency), Flu. (Fluency), Rel. (Relevance)]
>
> Analysis:
>
> - Adaptability to Independent Schemes: When prompted with the independent aspect schemes from Themis (UFEvalT), UFEvalT not only effectively executes the evaluation but also outperforms the original Themis baseline.
> - Instruction Following vs. Overfitting: The performance of UFEvalT proves that the model is actively following the definitions provided by the Themis scheme, rather than reverting to a memorized "FRABench style."
>
> **Conclusion:** This confirms that UFEval is not overfit to the FRABench. Its success on the independent Themis evaluation scheme demonstrates a robust capability to interpret and execute evaluation under a distinct evaluation scheme.

---

> > ### Comment · Reviewer_CEqP · 2025-11-23
> > **Response to Rebuttal**
> >
> > Thank you so much for the changes added to the paper. If I could give a 7 as a score, I would. Unfortunately, I'm still on the fence and would need experimental validation with another model than one from the Qwen family to be convinced to raise the score to 8.

---

> ### Author Response · Authors · 2025-11-30
> **Response to reviewer CEqP**
>
> Dear Reviewer CEqP,
>
> We sincerely thank you for the positive feedback and truly appreciate your willingness to reconsider the score based on further validation.
>
> To verify architectural generalization, we conducted experiments using LLaVA-Next (specifically llava-v1.6-mistral-7b-hf). Given that retraining on the full dataset requires significant resources and extensive time, we made our best effort to address this concern within the limited time by utilizing a smaller-scale dataset.
>
> (1). Experimental Setup
>
> To verify whether LLaVA-Next can evaluate unseen aspects after training on a distinct set of aspects, we constructed a training set by uniformly sampling 15k examples from each of the four tasks (NLG, IU, IG, and ITIG), totaling 60k samples. We then fine-tuned the LLaVA-Next model on this dataset using 32 A100 GPUs with a batch size of 64, while keeping all other training hyperparameters consistent with UFEval.
>
> (2). Experimental Results
>
> We evaluated both the original baseline (L-Next) and our fine-tuned model (L-Ours) on the FRA-OOD-H, presenting the results for unseen aspects on unseen tasks in Table 1 and seen aspects on unseen tasks in Table 2.
>
> ### Table 1: Evaluation of Unseen Aspects on Unseen Tasks
> | Method |Coverage|Appeal|Relevance|Precision|Format|Suitability|Simplification|Correctness|Difficulty|
> |--- |---|---|---|---|---|---|---|---|---|
> |**L-Next**|47.4|39.5|35.0|37.5|42.5|39.5|10.0|55.0|50.0
> |**L-Ours**  |**60.5**|**71.1**|**57.5**|**65.0**|**72.5**|**52.6**|**55.0**|**65.0**|**62.5**|
> |**Method**|**Translation Fidelity**|**Thoroughness**|**Expertise**|**Explainability**|**Terminology**|**Transparency**|**Interpretability**|**Completion**|**Representation**|
> |**L-Next**|46.2|35.9|62.5|40.0|42.5|45.0|32.5|27.5|52.5|
> |**L-Ours**  |**74.4**|**64.1**|**80.0**|**77.5**|**65.0**|**62.5**|**65.0**|**57.5**|**70.0**|
> |**Method**|**Riskiness**|**Scene Alignment**|**Negation**|**Universal**|**Comparison**|**Originality**|**Reasonableness**|**Fulfillment**|**Engagingness**|
> |**L-Next**|15.0|26.8|17.1|25.0|12.5|12.4|15.0|12.5|35.0|
> |**L-Ours**       |**60.0**|**44.5**|**36.6**|**37.5**|**47.5**|**57.5**|**54.5**|**47.5**|**75.0**|
>
> ### Table 2: Evaluation of Seen Aspects on Unseen Tasks
> | Method |Action Alignment|Action Consistency|Alignment|Attribute Alignment|Bias|Character Consistency|Clarity|Coherence|
> |--- |---|---|---|---|---|---|---|---|
> |**L-Next**|28.9|16.2|24.2|19.3|53.0|10.0|49.6|46.3|
> |**L-Ours**|**53.0**|**58.8**|**43.2**|**51.6**|**76.6**|**62.5**|**67.2**|**70.4**|
> |**Method**|**Color Alignment**|**Count Alignment**|**Cultural Bias**|**Ethnic Bias**|**Fidelity**|**Fluency**|**Privacy Violation**|
> |**L-Next**|54.1|18.5|21.1|60.5|35.7|49.3|60.0|
> |**L-Ours**|**62.0**|**25.9**|**63.7**|**84.7**|**53.7**|**78.5**|**80.0**|
> |**Method**|**Gender Bias**|**Grammaticality**|**Harmfulness**|**Image Coherence**|**Image Repetition**|**Object Alignment**|**Object Relationship**|
> |**L-Next**|77.5|45.6|50.2|14.4|12.5|18.5|22.4|
> |**L-Ours**|**86.0**|**70.6**|**68.2**|**51.1**|**55.0**|**38.9**|**45.5**|
> |**Method**|**Readability**|**Regional Bias**|**Scene Consistency**|**Semantic Consistency**|**Sentence Complexity**|**Shape Alignment**|**Spatial Alignment**|
> |**L-Next**|49.5|62.5|13.2|27.8|45.6|34.8|19.4|
> |**L-Ours**|**77.3**|**77.5**|**70.0**|**66.7**|**56.9**|**48.4**|**39.8**|
> |**Method**|**Spelling Accuracy**|**Structure Accuracy**|**Stylistic Consistency**|**Tense Consistency**|**Texture Alignment**|**Toxicity**|**Vocabulary Complexity**|
> |**L-Next**|49.4|32.9|17.5|43.6|13.0|52.5|43.8|
> |**L-Ours**|**68.4**|**70.9**|**56.2**|**71.8**|**37.3**|**75.0**|**60.6**|
>
> **Conclusion:** The results show that after training on a distinct set of aspects, LLaVA-Next generalizes to unseen aspects and transfers seen aspects to unseen tasks, resulting in performance improvements. This confirms that our findings are valid on other architectures.

---

### Official Review · Reviewer_1uwD · 2025-10-31

**Soundness:** 4
**Presentation:** 4
**Contribution:** 4
**Rating:** 8
**Confidence:** 4

**Summary:**

In this paper, the authors have developed a new evaluation dataset, FRABench, and used it to train the UFEval model, a unified evaluator of multimodal LLMs (MLLMs) open ended responses.

Their FRABench evaluation dataset contains 112 hierarchical, universal and task-specific eval aspects related to major MLLM tasks. The dataset contains 60,400 sample pairs and 325,000 evaluation labels, created using both human reviewers and GPT-4.0.

The authors argue that because these evaluation aspects are interconnected, it leads to better performance and enable the model to generalize to unseen aspects.

**Strengths:**

- They created a comprehensive, fine-grained dataset for MLLM tasks.
- The paper is well written, with detailed experiments, ablation studies, and comparisons on several public benchmarks.
- They show that FRABench can be used to fine-tune smaller 7B models, improving their performance to match larger 72B+ models for MLLMs tasks
- They show that UFEval can be used to automatically generate high-quality preference data.

**Weaknesses:**

- Heavy reliance on GPT-4o annotations. It is not shown how much they correlate with human labels.
- UFEval still underperform relative to the larger models.

**Questions:**

Could FRABench be used for retrieval, providing few-shot examples to improve the performance of bigger models via in-context learning?

---

> ### Author Response · Authors · 2025-11-21
> **Response to reviewer 1uwD**
>
> Dear Reviewer 1uwD,
>
> We appreciate your time and constructive feedback, which has been invaluable in improving our work. We also express our sincere gratitude for your positive assessment of our paper and for recognizing the value of our contributions.
>
> >W1: Heavy reliance on GPT-4o annotations. It is not shown how much they correlate with human labels.
>
> A: We sincerely appreciate the reviewer for raising this concern regarding the quality of the data annotated by GPT-4o. Below, we clarify our rationale for selecting GPT-4o and demonstrate its agreement with human annotators.
>
> **1. Justification for utilizing GPT-4o**
>
> Two primary factors drove our decision to adopt GPT-4o for data annotation:
>
> - **Adoption in Recent Literature:** Through our preliminary investigation, we found that many recent studies [1-3] utilize GPT-4o for data annotation. This preference is driven by findings that GPT-4o achieves high alignment with human annotators.
>
> - **Complexity and Scale:** Our evaluation framework involves 112 fine-grained aspects. Manually annotating a large-scale training set with such high dimensionality is extremely difficult. Moreover, manual annotation is highly susceptible to "fatigue errors" given the extensive number of evaluation criteria.
>
> **2. Agreement with Human Labels**
>
> Regarding the agreement between GPT-4o and human annotators, we clarify that this information was included in our original submission.
>
> As detailed in Table 2 of the manuscript, we evaluated the agreement between GPT-4o and human annotations. The results demonstrate that GPT-4o achieves an average agreement rate of 83%. We reproduce the results from Table 2 below for your convenience:
>
> Table 1: Agreement of GPT-4o with Human Annotations
> | Agreement | UAs. NLG |UAs. IU |UAs. IG | UAs. ITIG | TAs. NLG | TAs. IU|  TAs. IG | TAs. ITIG  |
> | ----- | ----- | ----- | ----- | ----- | ----- | ----- | ----- | ----- |
> | GPT-4o | 84.0 | 82.1 | 72.3 | 93.1 | 83.2 | 82.1 | 74.2 | 93.1 |
>
> >W2: UFEval still underperform relative to the larger models.
>
> A: We acknowledge that UFEval still lags behind proprietary models like GPT-4o and Claude due to the significant disparity in parameter scale. However, UFEval outperforms other task-specific open-source evaluators. Moving forward, we plan to leverage state-of-the-art models as foundational backbones to elevate performance further and progressively narrow this gap.
>
> >Q1: Could FRABench be used for retrieval, providing few-shot examples to improve the performance of bigger models via in-context learning?
>
> A: Yes. We agree that FRABench holds potential for retrieval-based few-shot learning.
>
> FRABench provides not only evaluation scores but also detailed reasoning traces. These reasoning processes act as explicit demonstrations that help larger models grasp the specific evaluation criteria, making it an effective strategy to enhance performance via In-Context Learning without the need for fine-tuning.
>
> [1] Llava-critic: Learning to evaluate multimodal models. (CVPR)
>
> [2] Prometheus-Vision: Vision-Language Model as a Judge for Fine-Grained Evaluation (ACL)
>
> [3] Halu-J: Critique-Based Hallucination Judge. (AAAI)

---

### Author Response · Authors · 2025-11-21
**Summary of the Rebuttal**

Dear Area Chair,

We sincerely appreciate your time and effort in managing the review process. We are writing to provide a concise summary of our efforts during the rebuttal process, recapping our core contributions, the positive feedback from reviewers, and the revisions made to the manuscript in response to reviewers' suggestions.

Throughout the review process, the reviewers not only provided insights that helped us further refine our work, but also recognized the contributions of our research and acknowledged the value of our work in the following aspects:

- This work tackles a **well-recognized bottleneck** (Reviewer B3KA) in MLLM research. By identifying the **limitations of current approaches**, it proposes a **compelling, unified vision for evaluation** (Reviewer B3KA). To realize this vision, it addresses the **critical lack of resources** (Reviewer vDBt) required for training unified evaluators.
- We introduced FRABench, a **comprehensive, fine-grained dataset** (Reviewer 1uwD). It features a **clear hierarchical taxonomy** (Reviewer CEqP) and represents a **valuable conceptual contribution** providing a framework that **could be widely adopted** (Reviewer B3KA).
- UFEval demonstrates **competitive and good performance** (Reviewers B3KA, vDBt) compared to baselines across diverse benchmarks, and the work presents **extensive experiments and ablation studies** (Reviewers vDBt, 1uwD).

Regarding the reviewers' concerns, we have addressed them through additional experiments and detailed clarifications. This effort led two reviewers to raise their scores, while another reviewer expressed a willingness to increase the score conditional on further experiments. **For clarity, we summarize the reviewers’ final feedback and score updates below:**

- **Reviewer 1uwD — Rating: 8, Confidence: 4**

  The reviewer did not provide further comments following our response.

- **Reviewer CEqP — Rating: 6, Confidence: 3 (Reply received: 24 Nov 2025, 04:20)**

  The reviewer responded: "*Thank you so much for the changes added to the paper. If I could give a 7 as a score, I would. Unfortunately, I'm still on the fence and would need experimental validation with another model than one from the Qwen family to be convinced to raise the score to 8.*" We subsequently conducted the experiments. Unfortunately, the rebuttal process concluded before the reviewer could review the response.

- **Reviewer B3KA — Rating: 4 $\to$ 6, Confidence: 3 (Reply received: 25 Nov 2025, 21:34)**

    The reviewer responded: “*Thanks for your detailed rebuttal. Most of my concerns are addressed, so I'd like to raise the score to 6.*”

- **Reviewer vDBt — Rating: 4 $\to$ 8, Confidence: 3  (Reply received: 24 Nov 2025, 10:44)**

    The reviewer responded: “*Thank you very much for the detailed response. I am satisfied with the rebuttal, and all my concerns have been fully addressed. I have raised my rating to 8. Good luck.*”

Finally, we have revised our manuscript in response to the reviewers' suggestions. **The modifications are summarized as follows:**

- **Elaboration on Aspect Tree Construction:** In **Section 3.1.2 (Line 159)**, we have added detailed information explaining the construction process of the aspect tree. (Addressing Reviewers vDBt and B3KA)

- **Integration of Failure Case Analysis:** In **Section 4.2 (Line 426)**, we have moved the analysis of underperforming aspects from the Appendix to the main text to provide deeper insights into model limitations. (Addressing Reviewers B3KA & CEqP)

- **Definition of Aspect:** In the **Abstract (Line 16)**, we have provided a clear definition of the term 'aspect'. (Addressing Reviewer CEqP)

- **Inclusion of New Baselines:** In **Section 4.3 (Line 378)**, we have incorporated the latest baselines, including Qwen3-VL, CIGEval, and Q-Eval, for a more comprehensive comparison. (Addressing Reviewer B3KA)

- **Clarification on Table 1 Comparability:** In **Section 1 (Line 56)**, we have added a note clarifying the data comparability across different evaluators. (Addressing Reviewer B3KA)

- **Distinction of Generalization Categories:** In **Section 4.2 (Line 414)**, we have explicitly distinguished between contextual generalization and novel aspect generalization to clarify our aspect generalization evaluation. (Addressing Reviewer vDBt)

- **Empirical Evidence for Aspects (Robustness & Distinctness):** In **Appendix E (Line 2754)**, we have provided specific Inter-Annotator Agreement scores and correlation heatmaps to validate the robustness and distinctness of our aspect taxonomy. (Addressing Reviewer vDBt)

We once again express our sincere gratitude to the reviewers for their constructive feedback, which has been instrumental in refining our work. We also deeply appreciate the Area Chair’s dedication and the additional time and effort invested in handling our submission.

Best regards,

The Authors

---

### Meta-Review · Area_Chair_zKbd · 2026-01-06

**Summary:**

Major concerns shared by the reviewers include:
1. All labels in the dataset are generated by GPT-4o, raising concerns about potential bias inherited from the model
2. Lacking failure modes analysis
3. The presentation requires improvement
Additional concerns raised by the reviewers include:
1. Limited evidence supporting generalizability across different backbone models
2. Lacking key controlled experiments

**Reviewer Concerns:**

The rebuttal adequately addresses all of the reviewers’ concerns, with the only exception of the generalizability across different backbone models. The authors acknowledge this limitation due to resource constraints and have clearly stated it in the revised paper.

**Reviewer Scores:**

Most reviewers are satisfied with the rebuttal and indicated that they would raise their scores without the need for further discussion.

---

### Decision · Program_Chairs · 2026-01-26

Accept (Oral)